

**Direct radiative effect of carbonaceous aerosols from crop residue**
**burning during the summer harvest season in East China**
Huan Yao[1], Yu Song[1,*], Mingxu Liu[1], Tingting Xu[1], Pin Du[1], Jianfeng Li[1], Yusheng Wu[1],
Min Hu[1], Tong Zhu[1]
[1] State Key Joint Laboratory of Environmental Simulation and Pollution Control,
Department of Environmental Science, Peking University, Beijing, China
* Corresponding author: songyu@pku.edu.cn
**Abstract**

9       The East China experiences extensive crop residue burning in fields during harvest

seasons. The direct radiative effect (DRE) of carbonaceous aerosols from crop residue
burning in June 2013 in East China was investigated using the Weather Research and
Forecasting Model coupled with Chemistry (WRF-Chem). Absorption of organic aerosol
(OA) in the presence of brown carbon (BrC) was considered using the parameterization
of Saleh et al. (2014), in which the imaginary part of BrC refractive index was a function
of the ratio of the black carbon (BC) and OA and wavelengths. The carbonaceous
emissions from crop fires were estimated using the Moderate Resolution Imaging
Spectroradiometer (MODIS) fire radiative power products with a localized crop
burning-sourced BC-to-organic carbon (OC) ratio emission ratio of 0.27. The simulation
results were evaluated with in situ measurements of fine particle ($PM_{2.5}$) chemical
components and meteorological observations. The aerosol optical depths were
comparable with MODIS detections. The BC and OC peak concentrations reached 34.3





µg m$^{-3}$ and 121.1 µg m$^{-3}$, of which the crop residue burning contributed 86% and 90%,
respectively. Correspondingly, the DREs of crop residue burning-sourced BC and BrC
(due to absorption) reached +20.16 W m$^{-2}$ and +7.17 W m$^{-2}$, respectively. On average,
during the harvest season, crop residue burning introduced a DRE of +0.39 W m$^{-2}$
throughout East China. We found that BrC absorption and BC introduced significant
positive DREs, +0.85 W m$^{-2}$ and +1.05 W m$^{-2}$, respectively. The BrC DRE due to
scattering was stronger (−1.1 W m$^{-2}$) than its DRE due to absorption. The sensitivity test
showed that the BrC DRE strongly depended on the absorptivity and BC-to-OA ratio
emission ratio from crop residue burning, and the volume mixing treatment could result
in a higher positive DRE compared to the core/shell treatment.
**Keywords:** Carbonaceous aerosols; direct radiative effect; crop residue burning; East
China
**1. Introduction**

Carbonaceous aerosols emitted from biomass burning contribute 42% and 74% of

global black carbon (BC) and organic carbon (OC) emissions, respectively (Bond, 2004),
playing an important role in the radiation budget system (Chung et al., 2012; Hobbs et al.,
1997; Jacobson, 2014). The Intergovernmental Panel on Climate Change (IPCC) Fifth
Assessment Report stated that BC from biomass burning introduced a global mean direct
radiative forcing (DRF) of approximately +0.2 (+0.03 to +0.4) W m$^{-2}$, while that of
organic aerosol (OA) from biomass burning was about the same magnitude with the
opposite sign (Bond et al., 2013; Stocker, 2014). DRF is a measure of the change in





direct radiative effect (DRE) relative to preindustrial conditions, defined as year 1750 by
the IPCC. Precise computing of short term DRE caused by carbonaceous aerosols is
primary and essential to aerosol DRF estimation, avoiding the large uncertainties in
estimations of preindustrial carbonaceous aerosol emissions (Bond et al., 2013). DRE,
namely the radiative effect due to aerosol–radiation interactions, could also be a more
exhaustive gauge for comparisons between models and observations (Heald et al., 2014).

Aside from the established sunlight-absorbing BC from imperfect combustion

sources (Chang et al., 1982), the other co-emitted organic carbonaceous aerosol was
found to have an absorptive component closely linked to biomass burning (Kirchstetter et
al., 2004; Lack et al., 2012), commonly termed brown carbon (BrC) (Andreae, 1995),
which contributed a positive mean radiative forcing of $+0.1$ W m$^{-2}$ to $+0.25$ W m$^{-2}$ by
absorption globally (Feng et al., 2013). BrC radiation absorption is characterized by a
strong dependence on wavelength, increasing sharply from the short visible to the
ultraviolet ranges (Andreae and Gelencsér, 2006; Bond, 2001). The light absorption of
BrC from different sources is also highly variable; and for biomass burning, the
temperature of the combustion process, moisture content, and fuel type can be factors,
thus complicating the treatment of BrC in models (Laskin et al., 2015). Therefore, studies
using constant optical parameters of BrC for climate forcing calculations would have
significant uncertainties (Feng et al., 2013; Wang et al., 2014). Recently, Saleh et al.
(2014) proposed that the absorptivity of BrC from biomass burning, both fresh and aged,
could be parameterized as a function of the BC-to-OA ratio. This parameterization has
been used to simulate the DRE of BrC from biomass or biofuel burning emissions



globally in several studies (Kodros et al., 2016; Kodros et al., 2015; Saleh et al., 2015).
Off-line models with discrepancies in physical parameterizations between chemical and
meteorological simulations could also produce some errors (Gu et al., 2006).
As a large agricultural country, China emits approximately 97 Gg BC and 463 Gg OC
annually from crop residue burning in fields, mainly concentrated during the harvest
season in East China (Lu et al., 2011; Zhang et al., 2008).The previous emission
estimations were primarily derived from provincial statistical data with coarse spatial and
temporal resolutions. Only one study focused on the DRE of crop burning-sourced
carbonaceous aerosol over China and thus the DRE uncertainties could be very large,
especially during harvest season in East China (Li et al., 2016).
In this study, the DRE of carbonaceous aerosol from crop residue burning in East
China was quantified using the online Weather Research and Forecasting Model coupled
with Chemistry (WRF-Chem), with high-resolution carbonaceous aerosols emissions
from crop fires. The BrC absorptivity and its variation with wavelength and BC-to-OA
ratio were considered. The simulation was conducted for the harvest season in June 2013.
**2. Methods and Data**
2.1. Model Configuration
The online coupled meteorology-chemistry model, WRF-Chem version 3.6.1 (Grell
et al., 2005), was used. Double-nested domains centered at 36.5° N, 115.52° E, were set
with the coarse domain divided into 51 × 59 grid cells of 75-km horizontal resolution and
the fine domain divided into 48 × 63 grid cells of 25-km resolution (Fig. 1). The 25



vertical layers from the ground level to the top pressure of 50 hPa were used for all grids.
The Yonsei University (YSU) boundary layer vertical diffusion scheme (Hong and
Dudhia, 2003) was adopted. The global atmospheric reanalysis data ERA-Interim
produced by the European Centre for Medium-Range Weather Forecasts (ECMWF) was
used as the initial meteorological fields and boundary conditions with 3-hourly surface
parameters and 6-hourly upper-air parameters (Dee et al., 2011). The meteorology fields
were initialized at the start of each model run, which covered 36 h with the first 12 h as a
spinup. The simulated time covered the entire month of June, which was the local harvest
season of the main crop (wheat), and the model was conducted from 26 May to minimize
the impact from initial conditions. The domain settings and configuration options are
presented in Table 1.
For gas-phase chemistry, we chose the Model for Ozone and Related chemical
Tracers version 4 (MOZART-4) mechanism (Emmons et al., 2010) extended with clearer
aromatic compounds and monoterpenes treatments (Knote et al., 2014). The aerosol
processes, such as coagulation and thermodynamic equilibrium, were treated using the
Model for Simulating Aerosol Interactions and Chemistry (MOSAIC) scheme (Zaveri et
al., 2008), in which four discrete size bins were distinguished by dry physical particle
diameters (0.039–0.156, 0.156–0.625, 0.625–2.5, and 2.5–10.0 μm). A simplified
parameterization for secondary organic aerosol (SOA) formation was also incorporated
into the model by using CO from anthropogenic or biomass burning sources as a proxy
for SOA precursors based on the observed ratio between SOA and CO in polluted regions
(Hodzic et al., 2010). In WRF-Chem, aqueous phase chemistry was closely associated




with indirect effect modeling and was not included in this study.
The DRE estimates were derived from the instantaneous shortwave flux changes
between different scenarios (further explained in section 2.4) at top of atmosphere (TOA)
for average cloudy skies (i.e., all-sky conditions), with only direct radiative feedback
considered. Taking advantage of the multiple scattering handling capability and taking
both computing speed and accuracy into consideration, the rapid radiative transfer model
(RRTMG) (Mlawer et al., 1997) was selected to simulate shortwave flux change. We
ignored the radiative effect of gaseous materials emitted by crop residue burning and
focused on the shortwave aerosol DRE.
Aerosol optical properties, including absorption efficiency, scattering efficiency, and
the asymmetry parameter, are necessary for aerosol radiative transfer calculations. In this
study, these three parameters were computed by the shell/core Mie theory for each bin
(Ackerman and Toon, 1981) and then determined by summation over all size bins (Fast
et al., 2006). The spherical shell/core configuration was selected for the calculations of
aerosol optical properties, in which the BC core is assumed to be coated with a
homogeneously mixed shell of other species. For each bin, the complex refractive index
of the shell was derived by volume averaging that of every shell species (Barnard et al.,
2010), except for the imaginary refractive index of OA, which was zero by default. In
this study, we adopted the Saleh et al. (2014) parameterization to calculate the BrC
absorptivity, which formulating the imaginary part of OA's refractive index $k_{OA}$ with the
ratio of BC and OA from biomass burning as following:



$$k_{OA,550} = 0.016 \log_{10}\left(\frac{BC}{OA}\right) + 0.04 \qquad (1)$$

$$\omega = \frac{0.21}{\left(\frac{BC}{OA}+0.07\right)} \qquad (2)$$

$$k_{OA} = k_{OA,550}\left(\frac{550}{\lambda}\right)^{\omega} \qquad (3)$$

The $k_{OA,\,550}$ is the imaginary part of OA's refractive index at wavelength ($\lambda$) of 550nm
and $\omega$ is the wavelength dependence of $k_{OA}$.
2.2. Emission Inventory
The crop residue burning emissions were derived based on fire radiative power (FRP)
from the Moderate Resolution Imaging Spectroradiometer (MODIS) products (Liu et al.,
2015). The FRP method could reduce emission uncertainties when compared to
traditional methods, in which multi-parameters that depend on local agricultural practices
were used. Moreover, the crop residue fires were often small and not captured by
MODIS burned area products (Roy et al., 2008). The emissions based on the FRP method
had 1-km and daily resolutions.
The BC and OC emission factors from crop fires in this study (1.98 g/kg and 0.54
g/kg, respectively) were set specifically for the winter wheat residue burning in East
China. It was averaged using published emission factors from winter wheat combustion
simulating experiments in the field or laboratory (Hays et al., 2005; Li et al., 2007;
Dhammapala et al., 2007; Turn et al., 1997). The BC-to-OC ratio from crop burning was
0.27, falling within the range of 0.20–0.32 observed during harvest seasons in East China
(Li et al., 2014; Yamaji et al., 2010; Yang et al., 2008).





The Multi-resolution Emission Inventory for China (MEIC, see www.meicmodel.org)
database was applied for China, and the Mosaic Asian Anthropogenic Emission
Inventory (MIX, see http://www.meicmodel.org/dataset-mix.html) database (Li et al.,
2015) was applied for the surrounding countries, including power plant, industrial,
residential, and vehicle emissions.
2.3. In Situ Measurements and Other Data
Fine particle ($PM_{2.5}$) chemical components were sampled and analyzed from May 30
to June 27, 2013 at the site (33°54′37″ N, 116°45′46″ E) in Suixi, Anhui Province, China,
close to vast stretches of wheat fields, the nearest of which was only 1 km away. There
were two sampling periods each day: from approximately 7:40 (GMT+8.0) to 18:00 and
from 18:40 to 7:00 the next morning. BC and OA were assessed by a thermal/optical
carbon analyzer (Sunset Laboratory, Tigard, OR, USA) with quartz-fiber filters. More
complete detail on sampling and analysis can be found in Li et al. (2014).
The MODIS Level-2 Atmospheric Aerosol Product (04_L2) data (Collection 6), at a
1-km daily resolution for June 2013, was used to evaluate the aerosol optical depth
(AOD) simulations, with the Deep Blue algorithms (Hsu et al., 2006) integrated with the
existing MODIS algorithm to retrieve AOD over the entire land area, including both dark
and bright surfaces.
2.4. Numerical Experiments
Seven parallel simulations were conducted to investigate the DRE of carbonaceous
aerosols from crop residue burning, as well as the effects of mixing state and BrC





absorption (Table 2). The default simulation, namely the BASE simulation, covered the
whole emissions and presumed BC cores coated by shells of other well-mixed aerosols.
The OA absorption was parameterized based on Saleh et al. (2014). The crop residue
burning DRE was estimated by the difference between the BASE and NOCB runs. To
compute the DRE from BC and OA from crop residue burning (i.e. NOBCCB and
NOOACB, respectively), we conducted two more parallel simulations without the
corresponding BC and OA emissions. Another simulation was performed by setting the
imaginary part of the OA refractive index to zero (NOBRC), to study the DRE values
caused by OA absorption.
**3. Results and Discussion**
3.1. Model Evaluation
The meteorological results in BASE simulation were evaluated by comparison with
the land-based station data in East China. Considering the extension of the inner domain
and the simulation period of June 2013, we chose the temperature and relative humidity at
2 m above ground surface (T2 and RH2, respectively) and the wind speed and direction at
10 m above ground (WS10 and WD10) from 221 matched stations. Statistical indices
(Table 3), including mean bias (MB), root-mean-square error (RMSE), fractional bias (FB),
fractional error (FE), and index of agreement (IOA), indicated that the model
well-simulated both temporal variations and spatial distributions of the four
meteorological items. The model well-reproduced the T2 and RH2, with IOAs of 0.92 and
0.87, respectively. The statistical indices of T2 had slightly better coincidence than those





of RH2, with the RMSE of RH2 reaching 13.93. There was a small underestimation
(−0.69%) of RH2, while WS10 was slightly overstimated (0.99 m/s). At three typical sites
(Fuyang, Yanzhou, and Xuzhou) corresponding to the three main districts affected by crop
fire (mentioned below), the model well captured the general temporal trends of T2 and
RH2, although the RH2 was slightly underestimated (Fig. S1), which might lead to small
differences in certain aerosol physical properties (Chapman et al., 2009; Xia et al., 2007).
In general, the simulation results were comparable to the meteorological observations.
The temporal variation of fire counts detected by MODIS in East China in June 2013
is shown in Fig. 2a. Approximately 97% of the fire counts occurred from 1–21 June, while
the fire counts decreased to < 200 per day after 21 Jun. Throughout the rest of this study,
we focus on the summer harvest from 1–21 June. The districts most affected by crop
residue fire were the southeastern Henan and central Anhui provinces from 1–8 June, and
then the northern Anhui province from 9–16 June, with most of the North China Plain
involved (Fig. 2b). Then, crop fires mainly occurred in the northern Jiangsu and eastern
Shandong provinces from 17–21 June, with diminishing fire counts. It is worth noting that
the longitude and latitude of the fire area gradually increased over time in three phases,
corresponding to the harvest time regulation, which was from inland to coastland and from
the south to the north, tightly tied to the summer air temperature differences between the
marine and terrestrial climate and low and high latitude, respectively.
The carbonaceous aerosols concentrations were well-reproduced when compared to
the measurements in Suixi (Fig. 3). BC and OC concentrations showed similar trends that
fluctuated smoothly with values < 10 μg m$^{-3}$ and 20 μg m$^{-3}$, respectively, and then the





concentrations began to increase on the night of 12 June, and reached a peak on the night
of 14 June night, with mean values of 34.3 μg m$^{-3}$ and 121.1 μg m$^{-3}$, respectively. The
peak value of OC was ~3−4 times that of BC, close to the BC-to-OC ratio of crop residue
burning emissions (0.27, in section 2.2), also indicating that the dominant source of
carbonaceous aerosols pollution was local biomass burning. During the severely polluted
period from 12–17 June, wheat residue burning contributed 68% and 73% of the BC and
OC concentrations, respectively, corresponding to the Positive Matrix Factorization results
(74.5% and 75.8%, respectively) in Li et al. (2014). The time variations of ammonium,
sulfate, and nitrate in PM$_{2.5}$ were also well-reproduced and had more fluctuation than that
of carbonaceous aerosols, indicating weaker correlation with the crop fires (Fig. S2).

The Suixi site was almost unaffected by the intensive fire counts in southeastern

Henan and central Anhui from 1–8 June, owing to the prevailing southeast wind, which
brought the pollutants to Henan, Shanxi, and northern Hebei Province (Fig. 4a). The peak
values of carbonaceous aerosols at the Suixi site were centralized around 12–16 June,
corresponding to the high fire counts in Northern Anhui during this period (Fig. 2). Most
of the North China Plain witnessed more than 15 μg m$^{-3}$ BC and 30 μg m$^{-3}$ OC due to the
local crop residue burning as well as the pollutants carried by the south wind. After 17
June, the main burning area moved east to the northern part of Jiangsu province, effecting
Shandong province and having less influence in Suixi. The main carbonaceous aerosols
polluted area during the summer harvest, with > 90% of the mass concentration
contributed by crop residue burning, also moved from south to north and from inland to
coastal areas (Fig. 4), corresponding to the fire counts distribution. Carbonaceous aerosols
increased rapidly in the evening at around 19:00–20:00 and reached peak values at dawn
(5:00–6:00), because of the relatively looser management of crop burning and weaker
boundary layer mixing at nighttime. After sunrise, the concentrations gradually decreased
as the fires slowly extinguished and the surface inversion coupled to layers aloft enhanced
vertical mixing (Cao et al., 2009).
The 550-nm AOD detected by MODIS was well-reproduced by WRF-Chem (Fig. 5a
and 5b), showing high values (above 1) in the North China Plain and Jinagsu, consistent
with the MODIS agricultural fire counts distribution during the summer harvest in Fig. 2.
Higher AODs in megacities, including Beijing, Shanghai, and Tianjin, might be
attributable to the increased sulfate and ammonium concentrations and scattering in
summer (Huang et al., 2015). We used linear interpolation between AOD at 400 nm and
600 nm to retrieve the AOD at 550 nm, as aerosol optical properties were computed only
at four wavelengths in the model (Nordmann et al., 2014). The MODIS AOD data around
23 sites were matched with the simulated AOD by hour, showing a normalized mean
deviation (NMD) of −16.1% and a correlation coefficient (R) of 0.52 (Fig. S3). This small
underestimation might be partly caused by the underestimation of the summer RH (Yoon
and Kim, 2006). Several studies have also noted that the MODIS retrieval AOD showed
high bias in ground-based measurements such as the Aerosol Robotic Network data
(Huang et al., 2015; Myhre et al., 2009; Zhao et al., 2013).
Aerosol absorption optical depth (AAOD) is an interactional outcome of three out of
the four factors that determine aerosol DRF (Bond et al., 2013; Schulz et al., 2006).
Similar patterns can be seen between the spatial distribution of 550-nm AAOD and the



carbonaceous aerosols concentration during the summer harvest (Fig. 5c and 5d),
especially at the junction of Henan, Anhui, Jiangsu, and Shandong. The short atmospheric
lifetimes of BC and BrC determined that the surface concentration was more restricted by
the map of emissions (Bond et al., 2013; Laskin et al., 2015; Zhuang et al., 2011), so the
serious pollution of carbonaceous aerosols and the higher AAOD might be traced from
local crop burning emissions. It is worth noting that we treat all-source OA as BrC, thus
amplifying the AAOD around the megacities of Beijing and Tianjin.
3.2. Direct Radiative Effect of Crop Residue Burning

Estimated by the difference between the BASE and NOCB simulations, a mean DRE

of $+0.39$ W m$^{-2}$ was introduced by crop residue burning at TOA in East China during the
summer harvest (Table 4), much higher than the previous open biomass burning DRE
estimation. The carbonaceous aerosols emitted from crop residue burning were the
dominant contributors to the DRE, including the climate warming agent BC and the
traditional cooling agent OA. Due to the lower BC-to-OA ratio compared to fossil fuel
and indoor biomass combustion, open biomass burning was previously simulated to
produce a cooling-to-neutral DRE (Abel et al., 2005; Chung et al., 2012; Myhre et al.,
2013). The incorporation of the BrC absorptivity scheme in this study might have led to
the positive crop residue burning DRE (Feng et al., 2013).

The DRE of BC from crop residue burning was calculated to be $+1.05$ W m$^{-2}$ at TOA

during the summer harvest based on the difference between the BASE and NOBCCB
simulations. This is higher than the DRE estimation from biomass burning BC ($+0.1$ W





278 m$^{-2}$ to +0.5 W m$^{-2}$) in East China for the summer of 2010 by Li et al. (2016), which used

279 the offline model with a coarse resolution. The emission inventories they used might

280 underestimate the BC emission from open biomass burning, especially during the harvest

281 season or in the burning zone, due to the traditional estimation methods and spatial

282 allocation rules (Lu et al., 2011). The external mixing state that they assumed would also

283 result in a lower and less accurate DRE than the core/shell treatment (Jacobson, 2001).

284 After dividing the DRE of BC from crop residue burning by the corresponding source

285 contribution to the BC mass concentration (13.5%, Table 5), our all-source BC DRE

286 estimate at TOA for the summer harvest of +7.8 W m$^{-2}$ was higher than the national

287 all-sky anthropogenic BC DRE for the summer of 2006 (+5 W m$^{-2}$) (Huang et al., 2015)

288 and in the range of the BC DRE in East China for the summer of 2008 (+5 W m$^{-2}$ to +15

289 W m$^{-2}$) (Gao et al., 2014). It was worth noting that these previous studies neglected the

290 crop residue burning emissions and adopted the volume mixing treatment, which would

291 definitely overestimate the BC DRE. So the missing of crop residue burning could cause

292 an underestimation of 7%–17% in anthropogenic BC DRE during the summer harvest.

293 Normalized DRE, defined by (Boucher and Anderson, 1995) (and first used in (Feichter

294 et al., 1997)) as the ratio of the forcing to the aerosol mass burden, was calculated to

295 isolate differences in the aerosol column burden from the differences in all other model

296 processes that lead to carbonaceous aerosols radiative forcing (Bond et al., 2013).

297 Normalized DRE with respect to the BC burden from crop residue burning was

298 +1,109.72 W g$^{-1}$ (Table 5), fell within the existing estimated global normalized DRF

299 ranges of +870 W g$^{-1}$ to +2730 W g$^{-1}$ (Bond et al., 2013; Ramanathan and Carmichael,





2008; Schulz et al., 2006).
By subtracting the TOA shortwave flux of NOOACB from that of BASE, we
obtained an OA DRE from crop residue burning of $-0.25$ W m$^{-2}$ in East China. The
normalized DRE of OA from crop residue burning, $-57.17$ W g$^{-1}$ (Table 5), was also
comparable to the existing estimates of $-24$ W g$^{-1}$ to $-198$ W g$^{-1}$ (Bond et al., 2013;
Ramanathan and Carmichael, 2008; Schulz et al., 2006). The DRE of OA from crop
residue burning, calculated by the shortwave flux differences between the BASE and
NOOACB simulations, was a combination of the positive BrC DRE ($+0.85$ W m$^{-2}$) due
to absorption and the negative DRE ($-1.1$ W m$^{-2}$) due to scattering. The former was
calculated by multiplying the total OA DRE from absorption (the shortwave flux
differences between the BASE and NOBRC simulations) by the crop residue burning
contribution to the total OA mass concentration of 30.3% (Table 5), which was in
agreement with the previous observed range of 24%– 67.5% at sites in East China (Fu et
al., 2012; Li et al., 2014). This was above the ranges of both the global annual mean BrC
DRE from absorption, of $+0.04$ to $+0.57$ W m$^{-2}$ (Feng et al., 2013; Saleh et al., 2015;
Wang et al., 2014), and the BrC DRE in East Asia for the spring of 2011, of $+0.1$ to $+0.2$
W m$^{-2}$ (Park et al., 2010). The BrC DRE from crop residue burning accounted for 81% of
the corresponding BC DRE, higher than the previous estimation range of 27%–70% (Lin
et al., 2014), indicating that BrC could be a dominant light-absorbing aerosol during the
summer harvest in East China.
Figure 6 illustrates that the high values of BC DRE (above $+3.0$ W m$^{-2}$) and BrC
DRE due to absorption (above $+1.5$ W m$^{-2}$) during the summer harvest mainly appeared



in the western Shandong, eastern Henan province, northern Anhui and northern Jiangsu
Provinces, similar to the spatial features of carbonaceous aerosol mass (Fig. 5d). The
hotspot was in the north of the intensive crop fire-affected area (Fig. 2b), as the dominant
southeastern wind in June transported the denser carbonaceous aerosols to the north
(section 3.1). With the carbonaceous aerosols mass concentration exceeding 35 μg m$^{-3}$,
the southeastern Henan and northern Jiangsu had the highest BC DRE above +5.0 W m$^{-2}$
and BrC DRE above +5.0 W m$^{-2}$ in our domain. The local DRE in the burning districts
during the crop residue burning periods could be higher than spatiotemporally averaged
estimates. Taking the Suixi site as an example, the mean crop residue burning
contributions to ambient BC and OA during the highest peaks (14–15 June) were 86%
and 90%, respectively. The corresponding mean DREs of crop residue burning-sourced
BC and BrC due to absorption reached +20.16 W m$^{-2}$ and +7.17 W m$^{-2}$, respectively.
3.3 Uncertainty
The DRE of carbonaceous aerosols were strongly dependent on the optical properties,
the uncertainties of which came from various factors, including complex refractive
indices, mixing state and the morphologies of the particles. Since this study was the first
attempt to use the BrC absorptivity parameterization of Saleh et al. (2014) in online
model, sensitivity experiments were conducted to investigate the response of BrC DRE
to the changes in the imaginary part of OA's refractive index ($k_{OA}$) and the key parameter,
namely the BC-to-OC emission ratio from crop residue burning. As the $k_{OA}$ was raised by
30% and 50%, the DRE of this source BRC due to absorption increased to +1.05 W m$^{-2}$
and +1.22 W m$^{-2}$ (Table S2). This DRE value was estimated to be +0.96 W m$^{-2}$ and





+0.81 W m$^{-2}$ (Table S2), when BC-to-OA ratio was set to 0.18 (Li et al., 2007) and 0.42
(Hays et al., 2005), respectively, with the OA emission factor from crop residue burning
in consistent with the standard simulation. The sensitivity test of BC-to-OC ratio could
account for the uncertainties introduced by variable residue burning conditions and OA
volatility (Kodros et al., 2015). Efforts are still needed to update the BC-to-OC ratio
localized ratio observation in China. These results indicated that the $k_{OA}$ and the
BC-to-OC emission ratio were critical for estimating BrC DRE. More details about
sensitivity test were presented in Table S1.
The sensitivity of BC mixing state to crop residue burning DRE was also tested by
changing the standard core/shell mixing to volume mixing, which assumed the particles
have a volume-averaged absorptivity and could lead to a higher absorption efficiency and
higher absorption coefficients than the former (Jacobson, 2000). In the volume mixing
treatment, crop residue burning was simulated to produce a mean DRE of +0.54 W m$^{-2}$
during the summer harvest (Table S2) The single-distribution core/shell assumption was
believed to be a better approximation of BC DRE (Bauer et al., 2013; Jacobson, 2001)
and more coated particles were observed in biomass burning aerosol (Schwarz et al.,
2008), so the widely-used volume mixing assumption could introduce an obvious DRE
discrepancy. The realistic carbonaceous aerosol mixing conditions are much more
various and complicated in time and space. For example, Peng et al. (2016) recently
reported that BC morphology varied from fractal particles to compact particles during
atmospheric aging, and BC in the two distinct stages revealed quite different absorption
characteristics and climatic effects. Therefore, the invariant core/shell assumption during



aging that we applied might overestimate the DRE of freshly emitted BC. This spherical
core/shell assumption might also amplify the absorption in cases in which the BC core
position is non-central (Adachi et al., 2010). The various moisture contents, as well as the
temperature conditions, also complicate the mixing state of carbonaceous aerosols and
the fraction and light absorptivity of BrC (Liu et al., 2013; Zhang et al., 2013). Moreover,
the lack of consideration of atmospheric processing of BrC, such as photobleaching
(Laskin et al., 2015), and the potential addition of nitrate groups (Jacobson, 1999), leads
to further uncertainties.

The high-resolution emission inventory based on the MODIS FRP used here may

add uncertainties to the carbonaceous aerosols mass concentrations and size distribution,
due to uncertainties arising from the MODIS detection resolution, FRP values, and the
per-fire-pixel Fire Radiative Energy (FRE) calculating method (Liu et al., 2015). The
simplified SOA formation scheme used in this study would also bring in uncertainties to
the OA concentration.

**4. Conclusion**

The DRE of carbonaceous aerosols from crop residue burning in June 2013 in

Eastern China was investigated using WRF-Chem. The BrC effective absorptivity
parameterization proposed by Saleh et al. (2014) was used. The carbonaceous aerosols
emissions from crop fires were estimated based on the MODIS FRP products, with the
localized BC-to-OC ratio from crop burning of 0.27. In situ observations conducted in
Suixi, Anhui Province, during the corresponding period were utilized to evaluate the





simulation. The WRF-Chem results well-captured the variation of carbonaceous aerosol
concentrations, showing peak pollution during the period from 12–17 June. The BC and
OC peak concentrations reached 34.3 μg m$^{-3}$ and 121.1 μg m$^{-3}$, of which the crop residue
burning contributed 86% and 90%, respectively. The simulation results also
well-reproduced the temperature and relative humidity from ground-based observations
and MODIS-detected AODs, although there was a slight overestimation of wind speed.
During the summer harvest in East China (1–21 June), similar patterns were found
among simulated AAOD, fire counts detected by MODIS, and carbonaceous aerosols
concentrations, with higher values in the junction of Shandong, Henan, Anhui, and
Jiangsu provinces, confirming that the crop residue burning was the dominant cause for
the high AAOD.

The DREs of crop residue burning-sourced BC and BrC due to absorption reached

+20.16 W m$^{-2}$ and +7.17 W m$^{-2}$ in Suixi. On average, during the concentrated harvest,
crop residue burning introduced a DRE of +0.39 W m$^{-2}$ throughout East China,
indicating that taking absorptive BrC into consideration caused the crop residue burning
DRE to become positive. The higher BC DRE (above +3.0 W m$^{-2}$) and BrC DRE due to
absorption (above +1.5 W m$^{-2}$) from crop residue burning during the concentrated
harvest mainly occurred in the North China Plain. BrC from crop residue burning as the
minor absorptive component brought about a significant positive DRE (+0.85 W m$^{-2}$),
accounting for 81% of the corresponding BC DRE (+1.05 W m$^{-2}$). The scattering
property of OA from crop residue burning was stronger (−1.1 W m$^{-2}$) than the absorptive
property, so the OA DRE was negative −0.25 W m$^{-2}$. The aerosol–radiation interaction





due to carbonaceous aerosols from crop residue burning in the summer harvest might
bring further effects on planetary boundary layer meteorology, turbulent kinetic energy,
cloud and precipitation (Liu et al., 2016; Huang et al., 2016; Wilcox et al., 2016). The
sensitivity test showed that the BrC DRE strongly depended on the absorptivity and
BC-to-OA ratio from crop residue burning, and the volume mixing treatment could result
in a higher positive DRE compared to the core/shell treatment. Several uncertainties
remain regarding the estimated DRE in this study, due to the mixing state and
morphology of the particles, burning conditions, and emission inventory. Continued
investigation of the mixing manner and ratio, the morphology and optical properties of
biomass burning aerosol, and their variation during the atmospheric aging process is still
required.
*Acknowledgements*. The MODIS Level-2 Atmospheric Aerosol Product (04_L2) data was
obtained from NASA L1 and Atmosphere Archive and Distribution System (LAADS),
USA. The ERA-Interim data was provided by the European Centre for Medium-Range
Weather Forecasts. This research was supported by National Natural Science Foundation
of China (41675142 and 41275155).



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

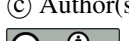



**Table Captions**


Table 1. WRF-Chem configuration options and settings.
Table 2. Descriptions of the parallel simulations.
Table 3. Statistical analyses of the simulated meteorological variables versus the
ground observations. MB, mean bias; RMSE, root-mean-square error; FB, fractional
bias; FE, fractional error; IOA, index of agreement.
Table 4. The DRE differences (W m$^{-2}$) between the cases at TOA during the summer
harvest (1–21 June) in 2013.
Table 5. The mass concentration contributions and normalized DRE of carbonaceous
aerosols from crop residue burning, and the DRE of OA from crop residue burning
due to absorption and scattering during the summer harvest (1–21 June) in 2013.





Table 1. WRF-Chem configuration options and settings

| Configuration options | |
| --- | --- |
| Radiation | RRTMG short- and longwave |
| Cumulus parameterization | New Grell Scheme (G3) |
| Land surface | Noah |
| Microphysics | Lin et al. |
| Photolysis | Fast-J |
| Gas chemistry | MOZART-4 |
| Aerosol chemistry | MOSAIC |
| Boundary layer | Yonsei University |
| Domain settings | |
| Horizontal grid | 52×60 (coarse) ; 49×64 (fine) |
| Grid spacing | 75 km×75 km (coarse); 25 km×25 km (fine) |
| Vertical layers | 25 |
| Projection | Lambert conformal conic |





Table 2. Descriptions of the main simulations.

| Simulation | Emission inventory | BC-to-OC ratio | OA absorptivity | Mixing state |
|---|---|---|---|---|
| BASE | Comprehensive | 0.27 | Saleh et al. (2014) | Shell/Core |
| NOCB | All without crop residue burning emissions | 0.27 | Saleh et al. (2014) | Shell/Core |
| NOBCCB | All without the BC emission from crop residue burning | 0.27 | Saleh et al. (2014) | Shell/Core |
| NOOACB | All without the OA emission from crop residue burning | 0.27 | Saleh et al. (2014) | Shell/Core |
| NOBRC | Comprehensive | 0.27 | None | Shell/Core |





Table 3. Statistical analyses of the simulated meteorological variables versus the
ground observations. MB, mean bias; RMSE, root-mean-square error; FB, fractional
bias; FE, fractional error; IOA, index of agreement.

| Index | MB[a] | RMSE[b] | FB[c] | FE[d] | IOA[e] |
|---|---|---|---|---|---|
| 2-m temperature (℃ ) | 0.26 | 2.72 | 0.01 | 0.09 | 0.92 |
| 2-m relative humidity (%) | –0.69 | 13.93 | –0.02 | 0.16 | 0.87 |
| 10-m wind speed (m/s) | 0.99 | 2.01 | 0.45 | 0.65 | 0.61 |
| 10-m wind direction (°) | 7.32 | 56.03 | | | |

[a]    $MB = \dfrac{1}{N}\sum_{1}^{N}(sim - obs)$
[b]    $RMSE = \sqrt{\sum_{1}^{N}(sim - obs)^2 / N}$
[c]    $FB = 2\sqrt{(sim_i - obs_i) / (sim_i + obs_i)} / N$
[d]    $FE = \sqrt{|sim_i - obs_i| / (sim_i + obs_i)^2} / N$
[e]    $IOA = 1 - \dfrac{N \times RMSE^2}{\sum_{i=1}^{N}(|obs_i - \overline{obs}| + |sim_i - \overline{obs}|)^2}$    , where the term sim and obs refer to the
simulated and observed meteorological values, respectively and N represents the
number of data pairs



Table 4. The DRE differences (W m$^{-2}$) between the cases at TOA during the summer
harvest (1–21 June) in 2013.

| BASE−NOCB | BASE−NOBCCB | BASE−NOOACB | BASE−NOBRC |
|---|---|---|---|
| +0.39 W m$^{-2}$ | +1.05 W m$^{-2}$ | –0.25 W m$^{-2}$ | +2.79 W m$^{-2}$ |





Table 5. The mass concentration contributions and normalized DRE of carbonaceous aerosols
from crop residue burning, and the DRE of OA from crop residue burning due to absorption
and scattering during the summer harvest (1–21 June) in 2013.

| Mass concentration contributions from crop residue burning (%)[a] | | Normalized DRE (W g$^{-1}$)[b] | | crop residue burning OA DRE (W m$^{-2}$) | |
|---|---|---|---|---|---|
| BC | OA | BC | OA | absorption [c] | scattering [d] |
| 13.5 | 30.3 | +1109.72 | −57.17 | +0.85 | −1.1 |

[a] The differences of BC or OA mass concentrations at surface between BASE and NOBCCB
or NOOACB divided by the corresponding BC or OA mass concentration at surface in BASE
simulation, respectively.
[b] The DRE of the crop residue burning sourced BC or OA (by subtracting the TOA shortwave
flux of NOBCCB or NOOACB from that of BASE, respectively) divided by the
corresponding crop residue burning sourced BC or OA mass column burden.
[c] By multiplying the total OA DRE due to absorption (BASE−NOBRC) with the mass
concentration contribution of OA from crop residue burning.
[d] The DRE of OA from crop residue burning (BASE−NOOACB) minus the part due to
absorption.



Figure Captions
Figure 1. Double-nested Weather Research and Forecasting Model (WRF) modeling
domains and topographic field (m); the sampling site (Suixi) is indicated by the red
dot.
Figure 2. (a) Time series of the fire counts detected by Moderate Resolution Imaging
Spectroradiometer (MODIS) in East China in June 2013. (b) Spatial distribution of
MODIS agricultural fire counts in East China in June 2013. The green, red and blue
dots represent the location of fire counts detected in 1–8 June, 9–16 June and 17–21
June, respectively.
Figure 3. Time series of the observed (dots) and simulated (line) (a) black carbon (BC)
and (b) organic carbon (OC) mass concentrations ($\mu$g m$^{-3}$) at the Suixi site.
Figure 4. Spatial distributions of (a) carbonaceous aerosols mass concentration ($\mu$g/m$^3$)
and (b) its contribution from crop residue burning (%) in the three typical hours (6:00)
during the summer harvest in June 2013.
Figure 5. Spatial distribution of mean (a) 550-nm aerosol optical depth observations
from MODIS, (b) 550-nm aerosol optical depth from WRF-Chem, (c) mean
absorption aerosol optical depth from WRF-Chem and (d) mean carbonaceous
aerosols concentration ($\mu$g m$^{-3}$) during the summer harvest. BASE run is shown.
Figure 6. Spatial distribution of (a) BC direct radiative effect (DRE) and (b) brown
carbon (BrC) DRE due to absorption from WRF-Chem during the summer harvest.





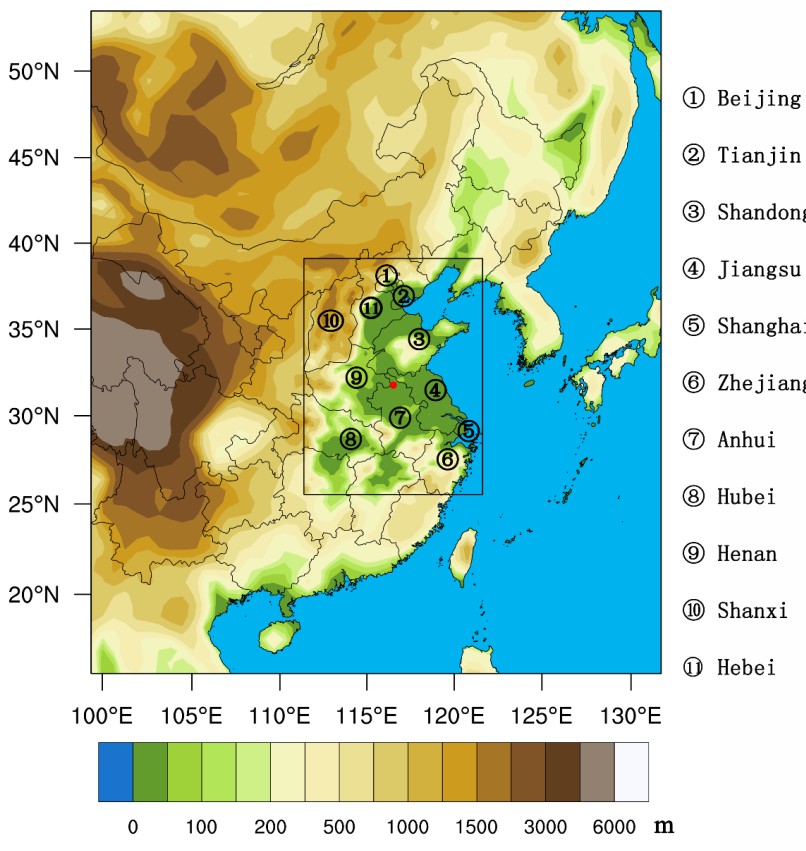


Figure 1. Double-nested Weather Research and Forecasting Model (WRF) modeling

domains and topographic field (m); the sampling site (Suixi) is indicated by the red

dot.



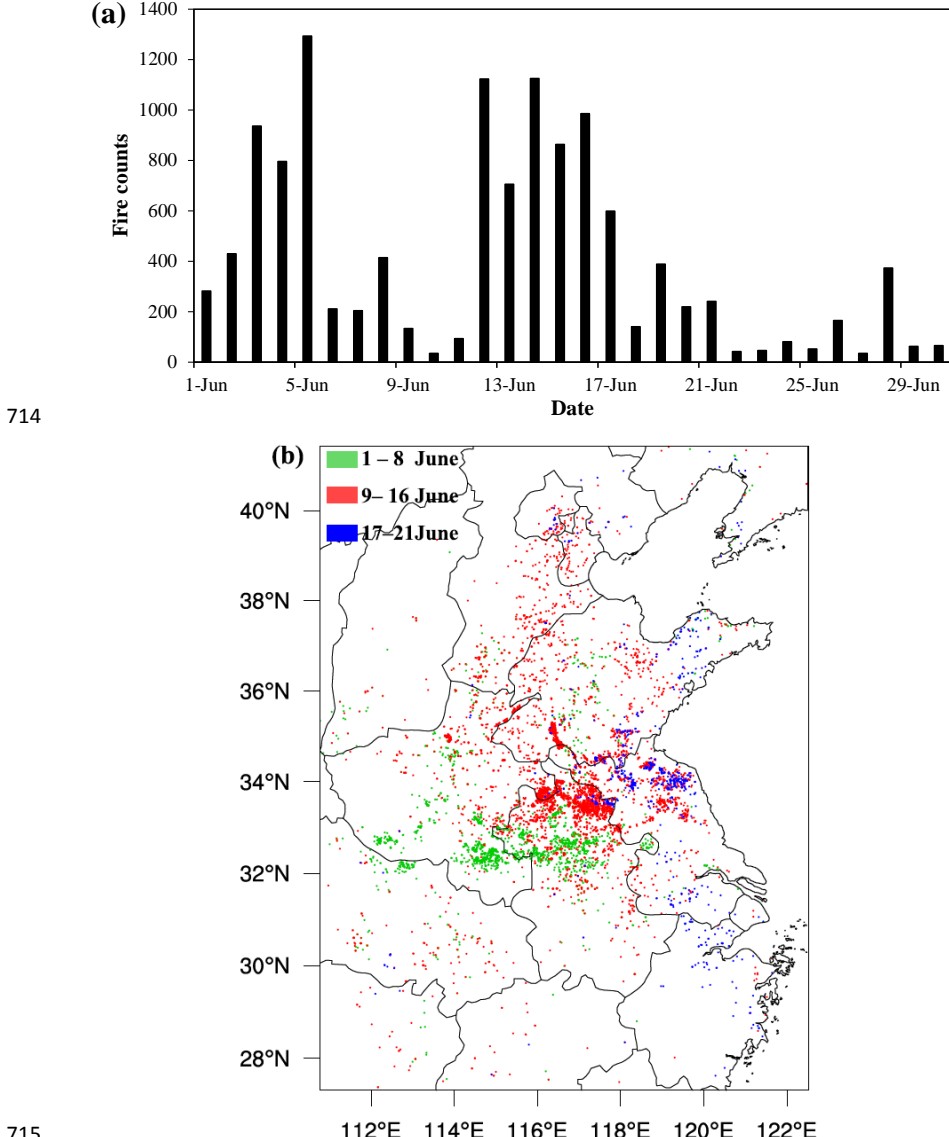



Figure 2. (a) Time series of the fire counts detected by Moderate Resolution Imaging

Spectroradiometer (MODIS) in East China in June 2013. (b) Spatial distribution of

MODIS agricultural fire counts in East China in June 2013. The green, red and blue

dots represent the location of fire counts detected in 1–8 June, 9–16 June and 17–21

June, respectively.

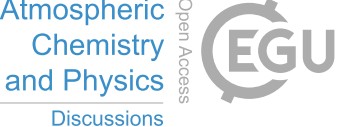



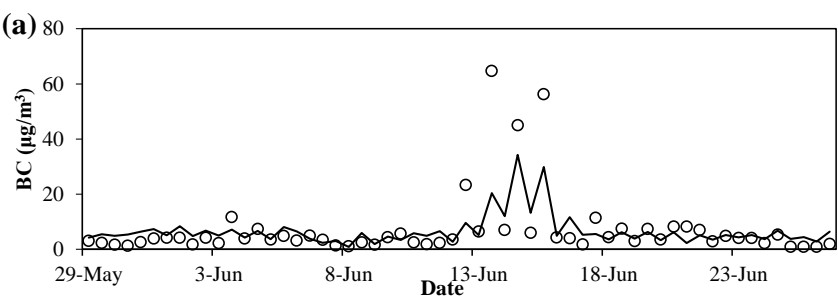



Figure 3. Time series of the observed (dots) and simulated (line) (a) black carbon (BC)
and (b) organic carbon (OC) mass concentrations ($\mu$g m$^{-3}$) at the Suixi site.



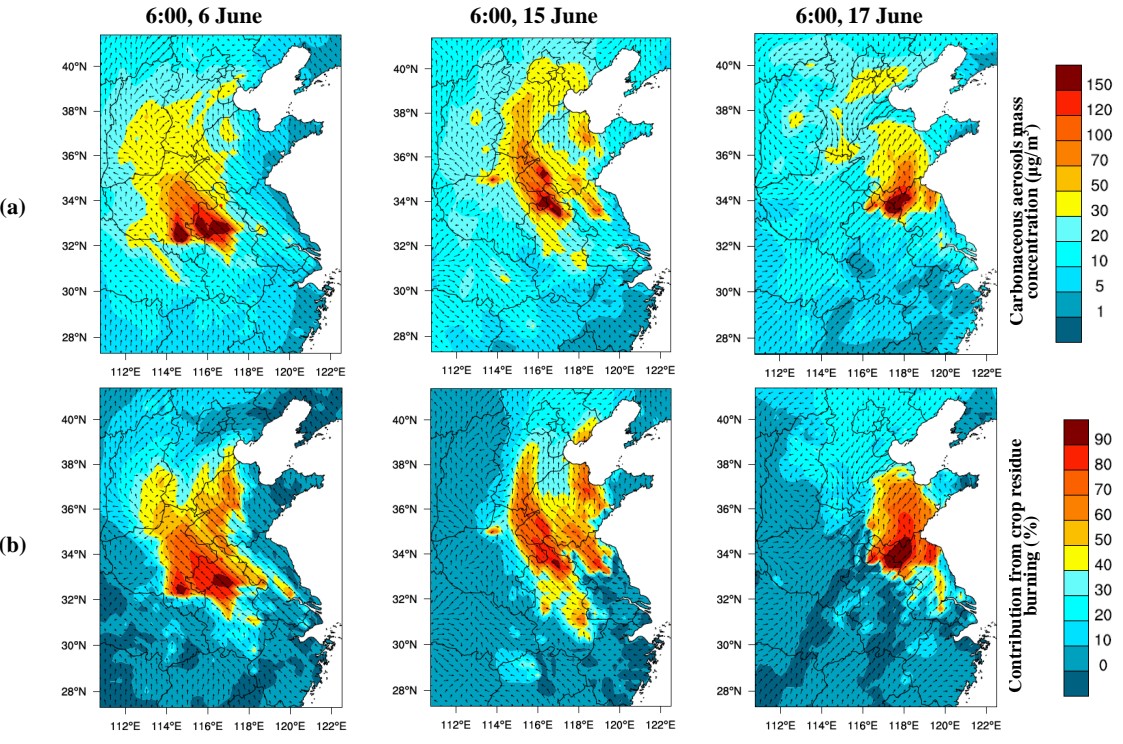

Figure 4. Spatial distributions of (a) carbonaceous aerosols mass concentration (µg/m³) and (b) its contribution from crop residue burning
(%) in the three typical hours (6:00) during the summer harvest in June 2013.




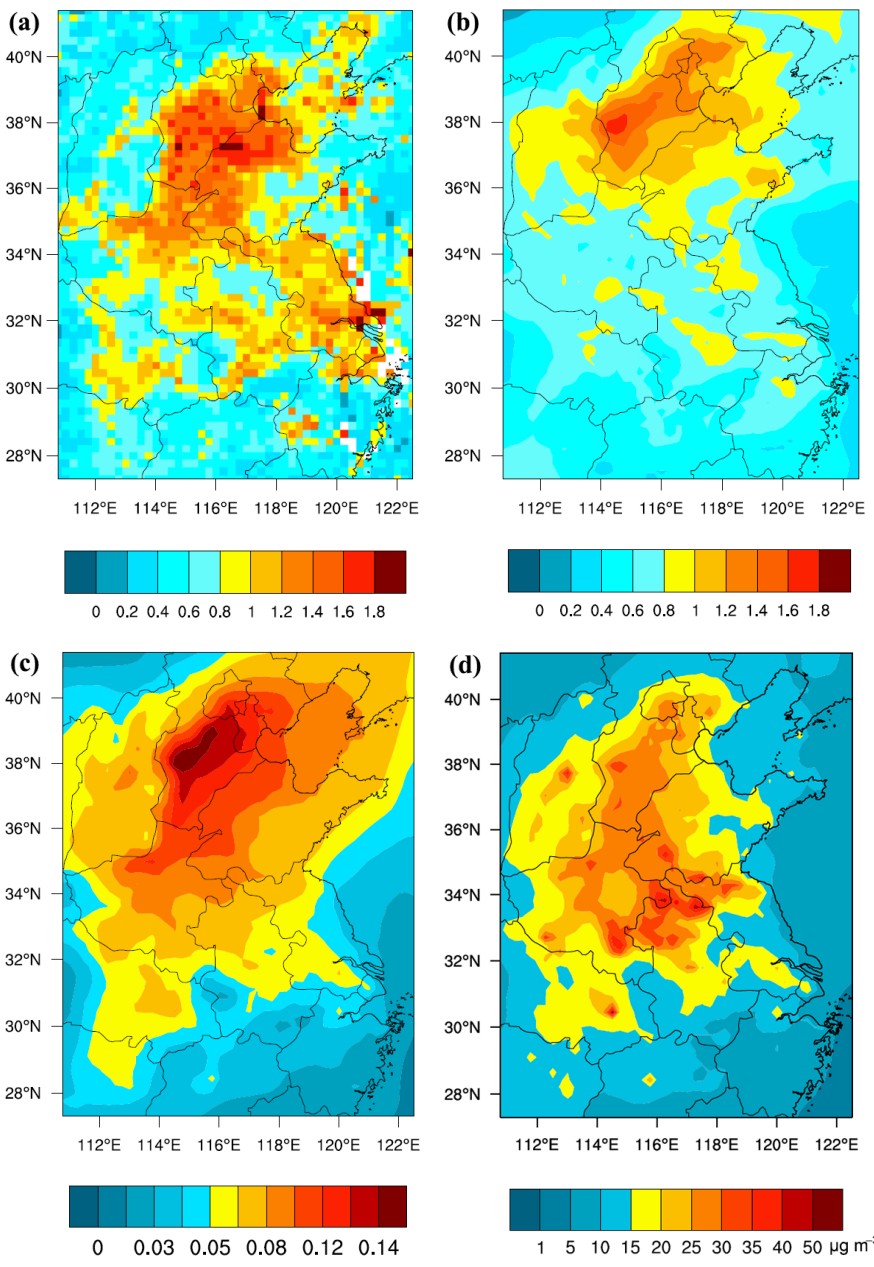

Figure 5. Spatial distribution of mean (a) 550-nm aerosol optical depth observations from MODIS, (b) 550-nm aerosol optical depth from WRF-Chem, (c) mean absorption aerosol optical depth from WRF-Chem and (d) mean carbonaceous aerosols concentration ($\mu$g m$^{-3}$) during the summer harvest. BASE run is shown.





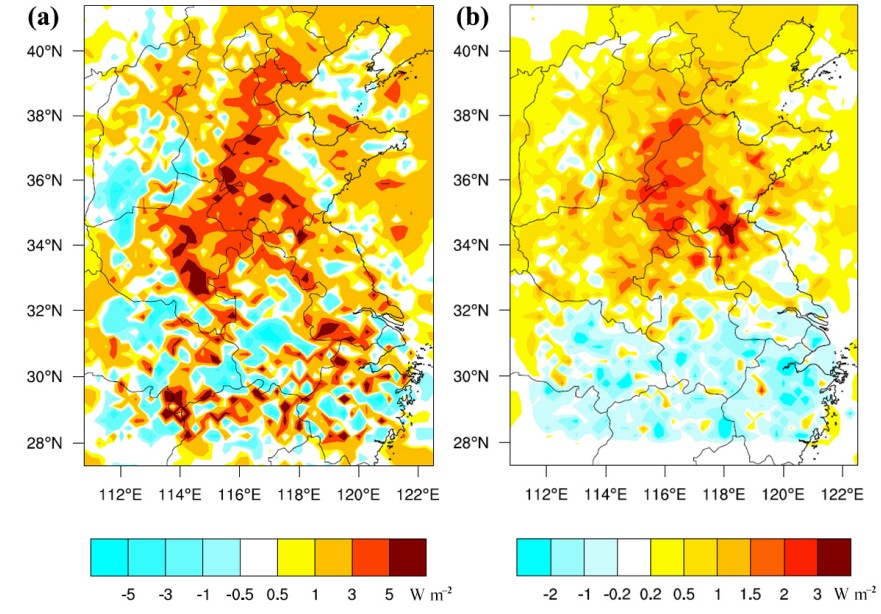

Figure 6. Spatial distribution of (a) BC direct radiative effect (DRE) and (b) brown

carbon (BrC) DRE due to absorption from WRF-Chem during the summer harvest.