# Peer review of "Direct radiative effect of carbonaceous aerosols from crop residue burning during the summer harvest season in East China"

_Atmospheric Chemistry and Physics, 2016_

## Referee Comment (RC1) · Anonymous Referee #2 · 6 Dec 2016

This is a well-written manuscript focusing on the timely subject of DRE of carbonaceous aerosols emitted by crop-residue burning in East China. I believe that the manuscript is suitable for publication in ACP. Below are a few comments: 1. Line 245: AOD calculations. Instead of doing linear interpolation to obtain AOD at 550, it is more appropriate to use a power-law fit – i.e. calculate the Angstrom Exponent from the other wavelengths. 2. Line 314: the cited studies present global maps of DRE due to BrC absorption. It would be informative to compare results not with the global means, but with the DRE in East China from those studies (as could be inferred from the maps). I expect this to actually yield a good comparison. 3. Figure 3: It would be useful to also show scatter plots of modeled vs observed with a 1:1 line. 4. Figure 6: Why are there

negative values for BC DRE and DRE due to BrC absorption? Those should be strictly positive.

---

## Referee Comment (RC2) · Anonymous Referee #1 · 6 Dec 2016

General Comments This paper uses the WRF-Chem model to derive estimates of direct radiative effects (DRE) over a region of Eastern China heavily affected by agricultural burning emissions, for the period June 2013. Model output is evaluated against some in-situ surface measurements over the period and MODIS satellite AOD products. The parameterisation of Saleh et al., 2014 is used to estimate the impact of including an absorbing portion of organic aerosol from biomass burning sources (brown carbon, BrC) on the DRE. Sensitivity simulations are carried out without biomass burning emissions, without the absorbing BrC component of OA, To my knowledge, this is the first study that includes an estimate of the radiative impact of BrC in WRF-Chem, an important first step in understanding the impacts of this currently highly uncertain

but potentially important aspect of OA. For that reason I think it would be suitable for publication with appropriate changes. However, a key issue I have with this paper is how the direct radiative effects are calculated. The authors do not have aerosol-cloud interactions, so there should be no indirect effects. However, the absorbing aerosol will still absorb radiation and affect cloud formation and dynamics (the semi-direct effect). What the authors currently describe as the DRE is really the sum of the DRE and SDRE. This issue needs to be appropriately tackled before the paper is suitable for publication. The paper is mostly well written, although the authors need to cite some more work from the field, and some improvements to English and structure of results section are needed to improve clarity (see details below). In addition, while uncertainties are discussed, there is no attempt to quantify them and there is no discussion of the statistical significance of their results (although I acknowledge it can be challenging to reach statistical significance over time short periods).

Specific Comments 1. Ln. 26: Here and elsewhere, the authors say BrC and BC introduced 'significant positive DREs', but do not calculate the statistical significance of these results. Unless the authors can prove this effect is statistically significant, I suggest avoiding use of the word "significant", as it commonly implies statistical significance in scientific writing.

2. Introduction: There are no references to any studies that have used WRF-Chem papers in the introduction. This is a severe oversight. The authors should discuss some other papers which use WRF-chem to estimate the DRE and other radiative effects of aerosol from biomass burning or other sources.

3. Ln. 66-67: The meaning of the sentence beginning "Off-line models with discrepancies..." is unclear in the context of the paragraph. Are the authors saying the previously cited articles use offline models? If so, please make this explicit. They should then describe how this can be improved with online models (e.g. WRF-chem), with appropriate references backing up this statement.

4. Ln 109-112. The Authors are using an incorrect definition of DRE here. The authors correctly state that only having direct radiative feedbacks and not aerosol-cloud interactions, there is no indirect effect in their simulations. However, by using an online model, the radiative impacts of absorbing aerosol will impact cloud formation, circulation and distribution in the model (i.e. there are semi-direct effects going on, indeed this is the advantage of using an online model over an offline one). Therefore, by evaluating all-sky TOA radiative fluxes, the authors are really presenting the combination of direct and semi-direct effects. A rough estimate of just the DRE can be calculated comparing the TOA clear-sky radiative fluxes between scenarios. A more rigorous calculation of direct (and semi-direct) would need double-radiation calls, with additional fluxes calculated without aerosol radiative interactions. Please see for example Ghan et al., 2012 and Archer-Nicholls et al., 2016 for further discussion on this.

5. Ln 123-128. This is phrased confusingly by first saying the refractive index of OA is 0, then immediately saying how the refractive index of OA is parameterized. I would suggest rephrasing as: "For each bin, the complex refractive index of the shell was derived by volume averaging that of every shell species (Barnard et al., 2010). By default, the imaginary refractive index of OA is zero. In this study, we adopted the Saleh et al. (2014) parameterization..."

6. Can the authors comment on how the Saleh parameterisation was developed, what its appropriate uses are, and how it has been used previously and tested? E.g. what data sources where used to derive it, from what emissions sources. This would help understanding of uncertainties associated with the Salah et al., 2014 parameterisation for those not familiar with it.

7. The authors switch between OC and OA a lot. Do they multiply OC from emissions/model output by a factor to give total OA? If so, please give the factor.

8. Section 2.2: Can the authors comment on how their emission inventory compares to other more commonly used biomass burning inventories over the region, in total mass

emissions for OC and BC.

9. Ln 210-14. It is not clear whether the fluctuations described are about the measurements or model. Please explicitly describe the observations first, then how the model behave in comparison. Please give some statistical measure of the model skill against these observations, as was done for the meteorological data.

10. Ln 234-239. What diurnal profile do the fire emissions have in your model? Are you saying the emissions from the fires are greater at night than during the day? I would expect the high surface concentrations at night are almost entirely due to the collapse of the nighttime boundary layer.

11. Ln. 267-8. Much higher that what previous DRE estimate? Please give hard numbers and references of previous estimates for this.

12. Section 3.2: are the radiative effects calculated over the whole month of June? Over the whole of the inner domain? Please be specific., these results really are just for a specific time and place and should not be interpreted as typical effects for the region. When comparing with results from other studies, are the comparisons over the same region over similar timeframes?

13. Ln 408-411. The authors discuss here that the aerosol will be bringing further effects on PBL, TKE, clouds and precipitation, but no attempt to present these changes is made. These changes will have an effect on the radiative balance (semi-direct effect) and should be documented.

Technical corrections: 1. Ln 9. Remove opening "The".

2. 76-77. No reference given for WRF-Chem (usually Grell et al., 2005).

3. Ln 70. Insert space in "Zhang et al., 2008).The previous..."

4. Ln 142. I think the emissions factors for BC and OC are the wrong way round (unless BC emissions are four times that of OC!)

5. Ln 168. The authors say seven parallel simulations were conducted, but only list 5 in Table 2.

6. Ln 208. Remove the word 'tightly', it is redundant in this sentence.

Figures and tables: Table 2. and 4. These are missing the scenario with volume mixing instead of core-shell. Please include as well. Figure 4. Is 6:00 local time or UTC? Why is this time typical? Please mark on the maps the location of the Suixi site for reference, and make the wind arrows larger and less dense so they are easier to see. Please confirm whether the arrows are surface wind fields. Figure 6. These are just BASE-NOBCCB and BASE-NOBRC runs? Please also show panels for the other scenarios. Over what timeframe are these calculated? I assume the blue areas in panel a. are due to changes in cloud fields. I would assume if plotting the actual direct effect from BC, or just the clear-sky fluxes, that figure would only have red shading.

References: Ghan, S. J., Liu, X., Easter, R. C., Zaveri, R., Rasch, P. J., Yoon, J.-H. and Eaton, B.: Toward a Minimal Representation of Aerosols in Climate Models: Comparative Decomposition of Aerosol Direct, Semidirect, and Indirect Radiative Forcing, J. Clim., 25, 6461–6476, doi:10.1175/JCLI-D-11-00650.1, 2012. Archer, S., Lowe, D., Schultz, D. M. and Mcfiggans, G.: Aerosol-radiation–cloud interactions in a regional coupled model : the effects of convective parameterisation and resolution, Atmos. Chem. Phys., 16, 5573–5594, doi:10.5194/acp-16-5573-2016, 2016.

---

## Author Comment (AC1) · 15 Mar 2017

**Response to Referee #2**

**″Direct radiative effect of carbonaceous aerosols from crop residue burning during the summer harvest season in East China (acp-2016-759)″**

*This is a well-written manuscript focusing on the timely subject of DRE of carbonaceous aerosols emitted by crop-residue burning in East China. I believe that the manuscript is suitable for publication in ACP. Below are a few comments:*

*1. Line 245: AOD calculations. Instead of doing linear interpolation to obtain AOD at 550, it is more appropriate to use a power-law fit – i.e. calculate the Angstrom Exponent from the other wavelengths.*

**Response:** Accepted. We recalculate the AOD and AAOD at 550nm using the power-law fit. Please see lines 341–343, Fig. 5b, 5c and S4.

**Revision in Lines 341–343 on Page 16:** "We calculated the AOD at 550 nm from that at 400 nm and 600 nm using the ångström exponent, as aerosol optical properties were computed only at four wavelengths in the model (Nordmann et al., 2014)."

**Revision in Fig. 5b, 5c and S4:**

[Figure]

[Figure]

Fig. 1. Spatial distribution of mean (a) 550-nm aerosol optical depth observations from MODIS, (b) 550-nm aerosol optical depth from WRF-Chem, (c) mean absorption aerosol optical depth from WRF-Chem and (d) mean carbonaceous aerosols concentration ($\mu g \, m^{-3}$)

[Figure]

Fig. S4. Scatterplots of simulated hourly AOD and corresponding MODIS AOD at 23 sites in June 2013. Normalized mean bias (NMB) and the correlation coefficient (R) are given in the scatterplot.

*2. Line 314: the cited studies present global maps of DRE due to BrC absorption. It would be informative to compare results not with the global means, but with the DRE in East China from those studies (as could be inferred from the maps). I expect this to actually yield a good comparison.*

**Response:** Accepted. We have compared our results with those studies in East China (Park et al., 2010; Feng et al., 2013) and reword that in the revised paper. Please see lines 422–428.

**Revision in Lines 422–428 on Page 20:** "The DRE of OA absorption during summer harvest in East China in our study was within the global annual mean DRE ranges of OA absorption, of +0.04 to +0.57 W m$^{-2}$ (Feng et al., 2013; Saleh et al., 2015; Wang et al., 2014), and higher than the estimates in East Asia for the spring of 2011, of +0.1 to +0.2 W m$^{-2}$ (Park et al., 2010). Feng et al. (2013) estimated an upper limit of annual mean DRE of OA absorption to be +0.25 to +0.5 W m$^{-2}$ in East China."

*3. Figure 3: It would be useful to also show scatter plots of modeled vs observed with a 1:1 line.*

**Response:** Accepted. The scatter plots of observations and modeling results of BC and OC with the calculated normalized mean bias and correlation coefficient have been added in Fig. 3.

**Revision in Fig. 3:**

[Figure]

[Figure]

[Figure]

Fig. 2. Time series of the observed (dots) and simulated (line) (a) black carbon (BC) and (b) organic carbon (OC) mass concentrations ($\mu$g m$^{-3}$) at the Suixi site. Scatterplots of simulated (c) BC and (d) OC mass concentrations ($\mu$g m$^{-3}$) and corresponding observed values. NMB and R represent normalized mean bias and correlation coefficient, respectively.

*4. Figure 6: Why are there negative values for BC DRE and DRE due to BrC absorption? Those should be strictly positive.*

**Response:** Accepted. There are no negative values for BC DRE and DRE due to BrC absorption in our revised paper, because we have adopted an igorous calculating method of DRE by adding double-radiation calls to radiation drivers, following the radiation diagnostic module of Ghan et al. (2012) and Archer-Nicholls et al. (2016) et al. (2016). Please see lines 402–410 and 433–440.

**Revision in Fig. 6:**

[Figure]

Figure 3. Spatial distribution of simulated direct radiative effect (DRE) introduced by (a) all aerosol from crop residue burning and (b)BC from crop-burning, (c) OA from crop burning, and (d) the absorbing component of OA from crop-burning emissions, calculated from WRF-Chem simulations during the summer harvest (1–21 June).

**Revision in Lines 402–410 on Page 19:** "Figure 6b illustrates that the high values of BC DRE (above +2.0 W m$^{-2}$) during the summer harvest mainly appeared in the western Shandong, Tianjin Municipality, eastern Henan province, northern Anhui and northern Jiangsu Provinces, similar to the spatial features of >20 µg m$^{-3}$ carbonaceous

aerosol mass concentration (Fig. 5d). The hotspot was in the north of the intensive crop fire-affected area (Fig. 2b), as the dominant southeastern wind in June transported carbonaceous aerosol to the north (section 3.1). With the carbonaceous aerosols mass concentration exceeding 30 $\mu g\ m^{-3}$, the junction of Anhui, Shandong, Henan and Hebei Provinces witnessed the highest BC DRE in our domain of over +3.0 W $m^{-2}$."

**Revision in Lines 433–440 on Page 21:** "Figure 6c and 6d show a negative DRE of OA ($< -0.2$ W $m^{-2}$) and positive DRE of OA absorption ($>0.2$ W $m^{-2}$) over the North China Plain, respectively. Like the spatiotemporally averaged estimates of OA DRE and its absorbing part ($-0.22$ W $m^{-2}$ and $+0.21$ W $m^{-2}$, respectively), the OA DREs in most grid cells have equal magnitude to the corresponding DRE of its absorption but show opposite sign. This implies that the negative DRE of OA scattering is roughly double the positive DRE of OA absorption in magnitude. The consideration of OA absorption therefore reduced the negative OA DRE estimates from crop burning by half."

**References:**

Archer-Nicholls, S., Lowe, D., Schultz, D. M., and McFiggans, G.: Aerosol‒radiation‒cloud interactions in a regional coupled model: the effects of convective parameterisation and resolution, Atmos. Chem. Phys., 16, 5573-5594, doi:10.5194/acp-16-5573-2016, 2016.

Feng, Y., Ramanathan, V., and Kotamarthi, V. R.: Brown carbon: a significant atmospheric absorber of solar radiation? Atmos. Chem. Phys., 13, 8607-8621, doi:10.5194/acp-13-8607-2013, 2013.

Ghan, S. J., Liu, X., Easter, R. C., Zaveri, R., Rasch, P. J., Yoon, J. H., and Eaton, B.: Toward a Minimal Representation of Aerosols in Climate Models: Comparative Decomposition of Aerosol Direct, Semidirect, and Indirect Radiative Forcing, J. Climate, 25, 6461-6476, doi:10.1175/JCLI-D-11-00650.1, 2012.

Nordmann, S., Cheng, Y. F., Carmichael, G. R., Yu, M., Denier Van Der Gon, H. A. C., Zhang, Q., Saide, P. E., Pöschl, U., Su, H., Birmili, W., and Wiedensohler, A.: Atmospheric black carbon and warming effects influenced by the source and absorption enhancement in central Europe, Atmos. Chem. Phys., 14, 12683-12699, doi:10.5194/acp-14-12683-2014, 2014.

Park, R. J., Kim, M. J., Jeong, J. I., Youn, D., and Kim, S.: A contribution of brown carbon aerosol to the aerosol light absorption and its radiative forcing in East Asia, Atmos. Environ., 44, 1414-1421, doi:10.1016/j.atmosenv.2010.01.042, 2010.

Saleh, R., Marks, M., Heo, J., Adams, P. J., Donahue, N. M., and Robinson, A. L.: Contribution of brown carbon and lensing to the direct radiative effect of carbonaceous aerosols from biomass and biofuel burning emissions, Journal of Geophysical Research: Atmospheres, n/a-n/a, doi:10.1002/2015JD023697-T, 2015.

Wang, X., Heald, C. L., Ridley, D. A., Schwarz, J. P., Spackman, J. R., Perring, A. E., Coe, H., Liu, D., and Clarke, A. D.: Exploiting simultaneous observational constraints on mass and absorption to estimate the global direct radiative forcing of black carbon and brown carbon, Atmos. Chem. Phys., 14, 10989-11010, doi:10.5194/acp-14-10989-2014, 2014.

---

## Author Comment (AC2) · 15 Mar 2017

**Response to Referee #1**

**"Direct radiative effect of carbonaceous aerosols from crop residue burning during the summer harvest season in East China (acp-2016-759)"**

*General Comments This paper uses the WRF-Chem model to derive estimates of direct radiative effects (DRE) over a region of Eastern China heavily affected by agricultural burning emissions, for the period June 2013. Model output is evaluated against some in-situ surface measurements over the period and MODIS satellite AOD products. The parameterisation of Saleh et al., 2014 is used to estimate the impact of including an absorbing portion of organic aerosol from biomass burning sources (brown carbon, BrC) on the DRE. Sensitivity simulations are carried out without biomass burning emissions, without the absorbing BrC component of OA, To my knowledge, this is the first study that includes an estimate of the radiative impact of BrC in WRF-Chem, an important first step in understanding the impacts of this currently highly uncertainbut potentially important aspect of OA. For that reason, I think it would be suitable for publication with appropriate changes. However, a key issue I have with this paper is how the direct radiative effects are calculated. The authors do not have aerosol-cloud interactions, so there should be no indirect effects. However, the absorbing aerosol will still absorb radiation and affect cloud formation and dynamics (the semi-direct effect). What the authors currently describe as the DRE is really the sum of the DRE and SDRE. This issue needs to be appropriately tackled before the paper is suitable for publication. The paper is mostly well written, although the authors need to cite some more work from the field, and some improvements to English and structure of results section are needed to improve clarity (see details below). In addition, while uncertainties are discussed, there is no attempt to quantify them and there is no discussion of the statistical significance of their results (although I acknowledge it can be challenging to reach statistical significance over time short periods).*

*Specific Comments 1. Ln. 26: Here and elsewhere, the authors say BrC and BC introduced 'significant positive DREs', but do not calculate the statistical significance of these results. Unless the authors can prove this effect is statistically significant, I suggest avoiding use of the word "significant", as it commonly implies statistical significance in scientific writing.*

**Response:** Accepted. We rewrote the description. Please see line 34–39.

**Revision in Lines 34–39 on Page 2:** "On average, the simulations showed that the crop residue burning introduced a net positive DRE of +0.14 W m$^{-2}$ at TOA

throughout East China, with BC from this source as the main heating contributor ($+0.79$ W m$^{-2}$). The OA DRE from crop burning ($-0.22$ W m$^{-2}$) was a combined effect of the positive DRE of absorption ($+0.21$ W m$^{-2}$) and a stronger negative DRE of scattering ($-0.43$ W m$^{-2}$)."

*2. Introduction: There are no references to any studies that have used WRF-Chem papers in the introduction. This is a severe oversight. The authors should discuss some other papers which use WRF-Chem to estimate the DRE and other radiative effects of aerosol from biomass burning or other sources.*

**Response:** Accepted. We added discussions about some other papers investigating the DRE of aerosol from biomass burning (Archer-Nicholls et al., 2016) and other resources (Huang et al., 2015; Zhao et al., 2013) in different regions with WRF-Chem. Please see lines 123–131.

**Revision in Lines 123–131 on Page 6:** "WRF-Chem contains the physics to simulate the aerosol DRE, but need extra radiation diagnostics to distinguish the DRE from other aerosol-radiation-cloud interactions. Both Huang et al. (2015) and Zhao et al. (2013) calculated the aerosol DRE in WRF-Chem by performing calculations of aerosol optical properties and radiative transfer multiple times with and without one aerosol component and its associated water. Following Ghan et al. (2012), Archer-Nicholls et al. (2016) calculated the DRE due to biomass burning aerosols in WRF-Chem by using double calls to the radiation driver to derive extra diagnostic variables with the refractive index of all aerosol species set to zero."

*3. Ln. 66-67: The meaning of the sentence beginning "Off-line models with discrepancies: : :" is unclear in the context of the paragraph. Are the authors saying the previously cited articles use offline models? If so, please make this explicit. They should then describe how this can be improved with online models (e.g. WRF-chem), with appropriate references backing up this statement.*

**Response:** Accepted. We reworded. Please see lines 115–123.

**Revision in Lines 115–123 on Pages 6:** "The offline models used in the previous studies investigating warming due to OA absorption (e.g., GEOS-Chem) probably have induced errors from the inconsistencies in space, time and physical parameterizations between the separated atmospheric meteorological and chemical transport components. These errors could be circumvented in online models by integrating the chemical modeling into the meteorology simulation. Online models, such as the Weather Research and Forecasting Model coupled with Chemistry (WRF-Chem) (Fast et al., 2006; Grell et al., 2005), could provide further insight to aerosol-cloud-radiation feedbacks, which are crucial for understanding climate change (Zhang, 2008), but ignored in offline models."

*4. Ln 109-112. The Authors are using an incorrect definition of DRE here. The authors correctly state that only having direct radiative feedbacks and not aerosol-cloud interactions, there is no indirect effect in their simulations. However, by using an online model, the radiative impacts of absorbing aerosol will impact cloud formation, circulation and distribution in the model (i.e. there are semi-direct effects going on, indeed this is the advantage of using an online model over an offline one). Therefore, by evaluating all-sky TOA radiative fluxes, the authors are really presenting the combination of direct and semi-direct effects. A rough estimate of just the DRE can be calculated comparing the TOA clear-sky radiative fluxes between scenarios. A more rigorous calculation of direct (and semi-direct) would need double-radiation calls, with additional fluxes calculated without aerosol radiative interactions. Please see for example Ghan et al., 2012 and Archer-Nicholls et al., 2016 for further discussion on this.*

**Response:** Accepted. We have adopted the rigorous calculation method to diagnose DRE by adding double-radiation calls to radiation drivers, following the radiation diagnostic module of Ghan et al. (2012) and Archer-Nicholls et al. (2016). All the DREs were updated and the description of this radiation diagnostic module could be seen in lines 167–182.

**Revision in Lines 167–182 on Page 8:** "To distinguish the aerosol effect on radiation budget directly by absorbing and scattering from other aerosol-radiation-cloud interactions, we added diagnostic calls to the radiation driver, following Archer-Nicholls et al. (2016) and Ghan et al. (2012). "Clean-sky" diagnostic variables (e.g. $SW_{cln}$), defined as what the net radiative fluxes at the top and bottom of the atmosphere would be if there were no aerosol in the column, were calculated by calling the radiation driver with the complex refractive index of all aerosol species set to zero. Thus, the clean-sky variables include the radiation scattering and absorbing effects of clouds, but ignore all aerosol radiation scattering and absorption. The DRE of all the aerosol species (ADRE) at the top of atmosphere (TOA) can be diagnosed by the difference of all-sky (including all the aerosol-radiation-cloud interactions) and clean-sky short wave irradiances at TOA:

$$ADRE = \left(SW_{TOA}^{\downarrow} - SW_{TOA}^{\uparrow}\right) - \left(SW_{TOA,cln}^{\downarrow} - SW_{TOA,cln}^{\uparrow}\right) \qquad (1)$$

Where $SW_{TOA}^{\downarrow}$ and $SW_{TOA}^{\uparrow}$ represent the short wave radiation fluxes in down and up direction at TOA, respectively. The DRE estimates of crop residue burning and the related carbonaceous aerosols were then determined from the ADRE differences between scenarios (further explained in section 2.4)"

*5. Ln 123-128. This is phrased confusingly by first saying the refractive index of OA is 0, then immediately saying how the refractive index of OA is parameterized. I would suggest rephrasing as: "For each bin, the complex refractive index of the shell was derived by volume averaging that of every shell species (Barnard et al., 2010). By*

*default, the imaginary refractive index of OA is zero. In this study, we adopted the Saleh et al. (2014) parameterization: : :"*

**Response:** Accepted. Please see lines 192–194.

**Revision in Lines 192–194 on Pages 9:** "For each bin, the complex refractive index of the shell was derived by volume averaging that of all non-BC species (Barnard et al., 2010). By default, the imaginary refractive index of OA is zero."

*6. Can the authors comment on how the Saleh parameterisation was developed, what its appropriate uses are, and how it has been used previously and tested? E.g. what data sources where used to derive it, from what emissions sources. This would help understanding of uncertainties associated with the Salah et al., 2014 parameterisation for those not familiar with it.*

**Response:** Accepted. Saleh's parameterization was derived from smog chamber experiments with different biofuels. It can be used to character the effective absorptivity of organic aerosols emitted from biomass and biofuel burning. This parameterization has been used to simulate DRE of BrC from biomass or biofuel burning emissions globally in several studies (Kodros et al., 2016; Kodros et al., 2015; Saleh et al., 2015). Please see lines 106–110 and 194–201.

**Revision in Lines 106–110 on Pages 5:** "Recently, Saleh et al. (2014) proposed that the absorptivity of OA from biomass burning, both fresh and aged, could be parameterized as a function of the BC-to-OA ratio. This parameterization has been used to simulate the DRE of OA absorption from biomass or biofuel burning emissions globally in several studies (Kodros et al., 2015; Saleh et al., 2015; Kodros et al., 2016)."

**Revision in Lines 194–201 on Pages 10:** "In this study, we adopted the Saleh et al. (2014) parameterization to calculate the OA absorptivity, based on smog chamber experiments for both fresh and chemically aged emissions from globally important fuels to characterize the effective absorptivity of organic aerosols as a function of the ratio of BC to OA. This parameterization has been incorporated into the 3-D global chemical transport model GEOS-Chem to calculate global direct radiative effect of carbonaceous aerosols emitted from biomass/biofuel burning (Saleh et al., 2015; Kodros et al., 2016; Kodros et al., 2015)."

*7. The authors switch between OC and OA a lot. Do they multiply OC from emissions/model output by a factor to give total OA? If so, please give the factor.*

**Response:** Accepted. The conversion relation between OC and OA was added in lines 226–230.

**Revision in Lines 226–230 on Pages 11:** "Note that when input into WRF-Chem, the OA emissions were calculated by multiplying OC emissions by a factor of 1.4 to account for the associated hydrogen and oxygen mass making up total OA. The simulated primary and secondary OC concentrations were calculated by dividing the simulated OA fields by factors of 1.4 and 1.8, respectively (Gilardoni et al., 2009)."

*8. Section 2.2: Can the authors comment on how their emission inventory compares to other more commonly used biomass burning inventories over the region, in total mass emissions for OC and BC.*

**Response:** Accepted. The emissions in this study were compared with GFEDv4.1 data (http://www.globalfiredata.org/index.html). Please see the lines 214–219 in the revised paper.

**Revision in Lines 214–219 on Pages 11:** "The total BC and OC emissions in this study were 4.3 Gg and 15.9 Gg during the month of June, 2013, close to the results by using agricultural statistics data (Huang et al., 2012), but almost ten times higher than those in GFEDv4.1 data (0.42 Gg and 1.32 Gg for BC and OC, respectively) (Randerson et al., 2012)."

*9. Ln 210-14. It is not clear whether the fluctuations described are about the measurements or model. Please explicitly describe the observations first, then how the model behaves in comparison. Please give some statistical measure of the model skill against these observations, as was done for the meteorological data.*

**Response:** Accepted. We have rewritten this part. Please see lines 299–307. The scatter plots of observations and modeling results of BC and OC with the calculated normalized mean bias and correlation coefficient have also been added in Fig. 3.

**Revision in Lines 299–307 on Pages 15:** "At the Suixi site, BC and OC surface concentration observations fluctuated smoothly with values < 10 $\mu$g m$^{-3}$ and 20 $\mu$g m$^{-3}$ in early June, respectively, and then began to increase on the night of 12 June, reaching peaks on the night of 13−15 June night with mean values of 55.3 $\mu$g m$^{-3}$ and 157.9 $\mu$g m$^{-3}$, respectively (Fig. 3a and 3b). The peak value of observed OC was about three times that of observed BC, close to the BC-to-OC ratio of crop residue burning emissions used in model (0.27, in section 2.2), indicating that the dominant source of carbonaceous aerosols pollution was local biomass burning. WRF-Chem well-reproduced the carbonaceous aerosols concentrations fluctuating trends (Fig. 3a and 3b), with the correlation coefficient of 0.74 (Fig. 3c and 3d)."

**Revision in Fig. 3c and 3d:**

[Figure]

Fig. 3. Scatterplots of simulated (c) BC and (d) OC mass concentrations (μg m$^{-3}$) and corresponding observed values. NMB and R represent normalized mean bias and correlation coefficient, respectively.

*10. Ln 234-239. What diurnal profile do the fire emissions have in your model? Are you saying the emissions from the fires are greater at night than during the day? I would expect the high surface concentrations at night are almost entirely due to the collapse of the nighttime boundary layer.*

**Response:** Accepted. The crop fire emissions diurnal profile was on basis of our previous peasant household survey results and on-the-spot inspection in East China, which showed that the field burning of crop residue mainly happened in the evening. At present, crop burning is forbidden in many areas of China, including in Anhui province. Wheat residues are burned when police supervision is lax, which is mainly during the nighttime (Li et al., 2014). In addition to the intensive emissions of crop fire, weaker boundary layer mixing during the nighttime might also be a contributing factor to the higher carbonaceous aerosols concentrations. Please see lines 218–219.

**Revision in Lines 218–219 on Pages 11:** "The diurnal allocation of the emissions was based on previous household surveys (Fig. S1, more detail could be found in the Supplement)."

**Revision in Fig. S1 in the Supplement:**

[Figure]

Fig. S1. The diurnal profile of crop-burning emissions in East China. The diurnal profile was derived from the results of household surveys in the countryside of East China in the summer of 2013. Face-to-face surveys were made in five counties (Dongping, Lixin, Shangqiu, Xiantao and Dongping) in four Provinces (Shandong, Anhui, Hubei and Henan), where crop residue burning was intensive. Through interviews with about 1500 farmer families, information on farming method including in-field crop burning was collected. The frequencies of firing time and fire durations were calculated for the diurnal profile of crop-burning emissions. The crop fires were also recorded during the observations at Suixi.

*11. Ln. 267-8. Much higher that what previous DRE estimate? Please give hard numbers and references of previous estimates for this.*

**Response:** Accepted. We rewrote this passage. Please see lines 364−370.

**Revision in Lines 364–370 on Pages 18:** "Calculated as the ADRE difference between the BASE and nCB simulations, a mean positive DRE of +0.14 W m$^{-2}$ was introduced by crop residue burning at TOA in East China during the summer harvest (Table 4). This is higher than previous cooling-to-neutral DRE estimations of open biomass burning (Archer-Nicholls et al., 2016; Abel et al., 2005; Chung et al., 2012; Myhre et al., 2013; Sakaeda et al., 2011), which might be mainly attributed to the incorporation of the OA absorptivity scheme of Saleh et al. (2014) in this study (Kodros et al., 2016; Kodros et al., 2015; Saleh et al., 2015)."

*12. Section 3.2: are the radiative effects calculated over the whole month of June? Over the whole of the inner domain? Please be specific., these results really are just for a specific time and place and should not be interpreted as typical effects for the*

*region. When comparing with results from other studies, are the comparisons over the same region over similar timeframes?*

**Response:** Accepted. The radiative effects are calculated over East China (inner domain) during the summer harvest, which was defined as the period of 1–21 June in section 3.1. Because about 97% of the fire counts occurred from 1–21 June, while the fire counts decreased to < 200 per day after 21 Jun. Please see lines 288–290.

As there are limited studies regarding the direct radiative effects by crop residue burning, comparisons are conducted between our result (+0.14 W m$^{-2}$) over East China and other DRE of open biomass burning in other regions, which were all cooling-to-neutral (Sakaeda et al., 2011; Abel et al., 2005; Archer-Nicholls et al., 2016; Chung et al., 2012; Myhre et al., 2013). When comparing BC radiative effects, we chose the results in summer over East China (referred from the map) in Li et al. (2016) and Gao et al. (2014). Please see lines 364–368 and 380–395.

**Revision in Lines 288–290 on Pages 14:** "Approximately 97% of the fire counts occurred from 1–21 June, while the fire counts decreased to < 200 per day thereafter. Throughout the rest of this study, we focus on the summer harvest period from 1–21 June."

**Revision in Lines 364–368 on Pages 18:** "Calculated as the ADRE difference between the BASE and nCB simulations, a mean positive DRE of +0.14 W m$^{-2}$ was introduced by crop residue burning at TOA in East China during the summer harvest (Table 4). This is higher than previous cooling-to-neutral DRE estimations of open biomass burning (Archer-Nicholls et al., 2016; Abel et al., 2005; Chung et al., 2012; Myhre et al., 2013; Sakaeda et al., 2011)."

**Revision in Lines 380–395 on Pages 18–19:** "This is higher than the DRE estimation from biomass burning BC (+0.1 W m$^{-2}$ to +0.5 W m$^{-2}$) in East China for the summer of 2010 by Li et al. (2016), which used an offline model with a coarse resolution. The emission inventories they used might have also underestimated BC emissions from open biomass burning, especially during the harvest season or in the burning zone, due to the traditional estimation methods and spatial allocation rules (Lu et al., 2011). The external mixing state that they assumed would also result in a lower and less accurate DRE than the core-shell treatment (Jacobson, 2001). After dividing the DRE of BC from crop residue burning by the corresponding source contribution to the BC mass concentration (17.6 %), our all-source BC DRE estimate at TOA for the summer harvest of +4.5 W m$^{-2}$ was lower than the national all-sky averaged anthropogenic BC DRE for the summer of 2006 (+5 W m$^{-2}$) (Huang et al., 2015) and BC DRE in East China for the summer of 2008 (+5 W m$^{-2}$ to +15 W m$^{-2}$) (Gao et al., 2014). It was worth noting that these previous studies adopted the volume mixing treatment, which would overestimate the BC DRE. Further, the neglect of crop residue burning emissions in Gao et al. (2014) might cause an underestimation."

*13. Ln 408-411. The authors discuss here that the aerosol will be bringing further effects on PBL, TKE, clouds and precipitation, but no attempt to present these changes is made. These changes will have an effect on the radiative balance (semi-direct effect) and should be documented.*

**Response:** Yes, the aerosol from crop residue burning would brought semi-direct effect due to its influence on the PBL, TKE and precipitation. A recent study (Huang et al., 2016) has investigated the impact of aerosol-radiation interactions due to crop-residue burning on the summer precipitation in China. Their results can help to understand the semi-direct effect of the crop-residue burning aerosols. As the evaluation of semi-direct aerosol effect remains large uncertainties, we will perform further researches on it in the future. In this study, we focused on the carbonaceous aerosols DRE from crop residue burning using a rigorous diagnosing method.

**Technical comments**

*1. Ln 9. Remove opening "The"*

**Response:** Accepted. Please see the line 15.

*2. 76-77. No reference given for WRF-Chem (usually Grell et al., 2005).*

**Response:** Accepted. Please see the line 121.

*3. Ln 70. Insert space in "Zhang et al., 2008). The previous: : :"*

**Response:** Accepted. Please see the line 63.

*4. Ln 142. I think the emissions factors for BC and OC are the wrong way round (unless BC emissions are four times that of OC!)*

**Response:** Accepted. Please see lines 220–221.

**Revision in Lines 220–221 on Page 11:** "The BC and OC emission factors from crop fires in this study (0.54 g/kg and 1.98 g/kg, respectively) were set specifically for winter wheat residue burning in East China."

*5. Ln 168. The authors say seven parallel simulations were conducted, but only list 5 in Table 2.*

**Response:** Accepted. Please see line 253.

*5. Ln 208. Remove the word 'tightly', it is redundant in this sentence.*

**Response:** Accepted. Please see line 292.

*Figures and tables: Table 2. and 4. These are missing the scenario with volume mixing instead of core-shell. Please include as well.*

**Response:** We moved the discussion of scenarios with volume mixing to section 3.3, following the other reviewer's suggestion. Because the volume mixing rule is not physical and cannot represent even limiting cases for atmospheric aerosols like core-shell and external mixing morphology. Considering the volume mixing state was still applied in several studies, we decided to use it to quantify the DRE uncertainty of BC mixing state by comparing the results between (BASE–nCB) and (VM_BASE–VM_nCB) (Table S1 and S2). Please see lines 457–461.

**Revision in Lines 457–461 on Page 22:** "The sensitivity of BC mixing state to crop residue burning DRE was also tested by changing the standard core-shell mixing rule to a volume mixing rule. In the volume mixing treatment, crop residue burning was simulated to produce a mean DRE of +0.23 $Wm^{-2}$ during the summer harvest (Table S2), 64% higher than the crop burning DRE in default runs (+0.14 $Wm^{-2}$)."

*Figure 4. Is 6:00 local time or UTC? Why is this time typical? Please mark on the maps the location of the Suixi site for reference, and make the wind arrows larger and less dense so they are easier to see. Please confirm whether the arrows are surface wind fields.*

**Response:** Accepted. We chose the local time 6:00 as the typical time because the carbonaceous aerosols mass concentration distribution at 6:00 could represent the most serious pollution condition of a day. The mass concentration of carbonaceous aerosols usually increased at night and reached peak values at dawn (5:00–6:00 in local time, GMT+8.0) in crop fire-affected area because of the relatively looser management of crop burning and weaker boundary layer mixing at nighttime. Please see lines 330–335.

We also redrew the Fig. 4, adjusting the wind arrows and site markers and confirming that the arrows represented surface wind fields.

**Revision in Lines 330–335 on Page 16:** "Carbonaceous aerosol surface concentrations increased rapidly in the evening at around 19:00–20:00 (GMT+8.0) and reached peak values at dawn (5:00–6:00, GMT+8.0), due to the relatively looser management of crop burning and weaker boundary layer mixing at nighttime. After sunrise, the concentrations gradually decreased as the fires slowly extinguished and the surface inversion coupled to layers aloft enhanced vertical mixing (Cao et al., 2009)."

**Revision in Fig. 4:**

[Figure]

Fig. 1. Spatial distributions of (a) carbonaceous aerosols mass concentration (μg/m$^3$) at ground level (20 m) and (b) its contribution from crop residue burning (%) in the three typical hours (6:00, GMT+8.0) during the summer harvest (1–21 June) in June 2013. The location of the sampling site (Suixi) is indicated by the black dot. The arrows represent the surface wind fields.

*Figure 6. These are just BASE-NOBCCB and BASE-NOBRC runs? Please also show panels for the other scenarios. Over what timeframe are these calculated? I assume the blue areas in panel a. are due to changes in cloud fields. I would assume if plotting the actual direct effect from BC, or just the clear-sky fluxes, that figure would only have red shading.*

**Response:** Accepted. Spatial distribution of simulated direct radiative effect (DRE) introduced by crop residue burning and crop burning-sourced OA had been added in Fig. 6 in addition to that of BC and BrC. Please see Fig. 6a and 6c. The corresponding discussion could be seen in lines 370–376 and 433–440.

The timeframes were the summer harvest, which was specified as 1–21 June in 2013. Because about 97% of the fire counts occurred from 1–21 June, while the fire counts decreased to < 200 per day after 21 Jun (lines 271–273).

There are no negative values for BC DRE and DRE due to BrC absorption in our revised paper as the new DRE diagnosing method was adopted. Please see Fig. 6b and 6d.

**Revision in Lines 370–376 on Page 18:** "The spatial distribution of crop residue burning DRE (Fig. 6a) shows similar patterns to that of the mean carbonaceous aerosols concentration, providing further evidences that the carbonaceous aerosols emitted from crop residue burning were the dominant contributors to the DRE. Positive DRE values mainly appeared in the North China Plain and higher ones (more than 0.5 W m$^{-2}$) were in eastern Henan, southwestern Shandong, northern Jiangsu and northern Anhui Province."

**Revision in Lines 433–440 on Page 21:** "Figure 6c and 6d show a negative DRE of OA (< –0.2 W m$^{-2}$) and positive DRE of OA absorption (>0.2 W m$^{-2}$) over the North China Plain, respectively. Like the spatiotemporally averaged estimates of OA DRE and its absorbing part (–0.22 W m$^{-2}$ and +0.21 W m$^{-2}$, respectively), the OA DREs in most grid cells have equal magnitude to the corresponding DRE of its absorption but show opposite sign. This implies that the negative DRE of OA scattering is roughly double the positive DRE of OA absorption in magnitude. The consideration of OA absorption therefore reduced the negative OA DRE estimates from crop burning by half."

**Revision in Fig. 6:**

[Figure]

Fig. 2. Spatial distribution of simulated direct radiative effect (DRE) introduced by (a) all aerosol from crop residue burning and (b)BC from crop-burning, (c) OA from crop burning, and (d) the absorbing component of OA from crop-burning emissions, calculated from WRF-Chem simulations during the summer harvest (1–21 June).

**References:**

Archer-Nicholls, S., Lowe, D., Schultz, D. M., and McFiggans, G.: Aerosol–radiation–cloud interactions in a regional coupled model: the effects of convective parameterisation and resolution, Atmos. Chem. Phys., 16, 5573-5594, doi:10.5194/acp-16-5573-2016, 2016.

Fast, J. D., Gustafson, W. I., Easter, R. C., Zaveri, R. A., Barnard, J. C., Chapman, E. G., Grell, G. A., and Peckham, S. E.: Evolution of ozone, particulates, and aerosol direct radiative forcing in the vicinity of Houston using a fully coupled meteorology-chemistry-aerosol model, J. Geophys. Res., 111, doi:10.1029/2005JD006721, 2006.

Ghan, S. J., Liu, X., Easter, R. C., Zaveri, R., Rasch, P. J., Yoon, J. H., and Eaton, B.: Toward a Minimal Representation of Aerosols in Climate Models: Comparative Decomposition of Aerosol Direct, Semidirect, and Indirect Radiative Forcing, J. Climate, 25, 6461-6476, doi:10.1175/JCLI-D-11-00650.1, 2012.

Grell, G. A., Peckham, S. E., Schmitz, R., McKeen, S. A., Frost, G., Skamarock, W. C., and Eder, B.: Fully coupled "online" chemistry within the WRF model, Atmos. Environ., 39, 6957-6975, doi:10.1016/j.atmosenv.2005.04.027, 2005.

Huang, X., Song, Y., Zhao, C., Cai, X., Zhang, H., and Zhu, T.: Direct radiative effect by multicomponent aerosol over China, J. Climate, doi:10.1175/JCLI-D-14-00365.1, 2015.

Zhang, Y.: Online-coupled meteorology and chemistry models: history, current status, and outlook, Atmos. Chem. Phys., 8, 2895-2932, 2008.

Zhao, C., Ruby Leung, L., Easter, R., Hand, J., and Avise, J.: Characterization of speciated aerosol direct radiative forcing over California, Journal of Geophysical Research: Atmospheres, 118, 2372-2388, doi:10.1029/2012JD018364, 2013.